# First Report of a Patient with MPS Type VII, Due to Novel Mutations in *GUSB*, Who Underwent Enzyme Replacement and Then Hematopoietic Stem Cell Transplantation

**DOI:** 10.3390/ijms20215345

**Published:** 2019-10-28

**Authors:** Patricia Dubot, Frédérique Sabourdy, Geneviève Plat, Charlotte Jubert, Claude Cancès, Pierre Broué, Guy Touati, Thierry Levade

**Affiliations:** 1Laboratoire de Biochimie Métabolique, Centre de Référence en Maladies Héréditaires du Métabolisme, Institut Fédératif de Biologie, CHU de Toulouse, 31059 Toulouse cedex 9, France; patricia.dubot@inserm.fr (P.D.); sabourdy.f@chu-toulouse.fr (F.S.); 2INSERM UMR1037, CRCT (Cancer Research Center of Toulouse), Université Paul Sabatier, 31037 Toulouse, France; 3Service d’Hématologie Pédiatrique, CHU de Toulouse, 31058 Toulouse, France; plat.g@chu-toulouse.fr; 4Service d’Hématologie Pédiatrique, CHU de Bordeaux, 33076 Bordeaux, France; charlotte.jubert@chu-bordeaux.fr; 5Hôpital des Enfants, Centre de Référence en Maladies Héréditaires du Métabolisme, CHU de Toulouse, 31059 Toulouse, France; cances.c@chu-toulouse.fr (C.C.); broue.p@chu-toulouse.fr (P.B.); touati.g@chu-toulouse.fr (G.T.)

**Keywords:** Mucopolysaccharidosis, enzyme replacement therapy, transplantation, glycosaminoglycan, glucuronidase, Sly syndrome

## Abstract

We report the case of a boy who was diagnosed with mucopolysaccharidosis (MPS) VII at two weeks of age. He harbored three missense β-glucuronidase (*GUSB)* variations in exon 3: two novel, c.422A>C and c.424C>T, inherited from his mother, and the rather common c.526C>T, inherited from his father. Expression of these variations in transfected HEK293T cells demonstrated that the double mutation c.422A>C;424C>T reduces β-glucuronidase enzyme activity. Enzyme replacement therapy (ERT), using UX003 (vestronidase alfa), was started at four months of age, followed by a hematopoietic stem cell allograft transplantation (HSCT) at 13 months of age. ERT was well tolerated and attenuated visceromegaly and skin infiltration. After a severe skin and gut graft-versus-host disease, ERT was stopped six months after HSCT. The last follow-up examination (at the age of four years) revealed a normal psychomotor development, stabilized growth curve, no hepatosplenomegaly, and no other organ involvement. Intriguingly, enzyme activity had normalized in leukocytes but remained low in plasma. This case report illustrates: (i) The need for an early diagnosis of MPS, and (ii) the possible benefit of a very early enzymatic and/or cellular therapy in this rare form of lysosomal storage disease.

## 1. Introduction

Mucopolysaccharidosis (MPS) type VII or Sly disease is an autosomal recessive (MIM 253220) disease resulting from mutations of the *GUSB* gene, encoding β-glucuronidase (GUSB), a lysosomal enzyme (EC 3.2.1.31) involved in the degradation of glycosaminoglycans (GAGs) [1]. This lysosomal storage disorder is one of the rarest MPS, with a birth prevalence varying from 0.02 to 0.24 per 100,000 live births [2,3]. The deficiency of the enzymatic activity results in the accumulation of undegraded GAGs chondroitin sulfate (CS), dermatan sulfate (DS), and heparan sulfate (HS) in multiple organs, plasma, and urine. Classically, patients present with hepatosplenomegaly, skeletal involvement, and neurological deterioration; non-immune hydrops fetalis is commonly observed in the most severe forms [4]. However, a broad range of clinical phenotypes is described, ranging from an attenuated to a severe form, depending on the extent of neurological involvement. A recent survey indicated that half of the patients die before the age of one [4].

Currently, in addition to supportive treatment, there are two specific treatments available that aim to reduce the GAGs’ accumulation: enzyme replacement therapy (ERT) and hematopoietic stem cell transplantation (HSCT). For ERT, a recombinant form of human GUSB (vestronidase alfa) has been recently developed and used successfully [5], allowing a reduction of urinary GAGs and an improvement of the organomegaly [6]. However, the intravenously injected rhGUSB does not cross the blood-brain barrier and has no effect on neurological signs, while HSCT can bring the enzyme into the brain via its secretion from donor-derived microglial cells and prevent or slow the neurological deterioration [7]. As learnt from other MPS types, this procedure should be performed at an early stage of the disease in the absence of preexisting neurological damage.

Here, we report the case of a boy who was diagnosed very early with MPS VII and was subsequently treated first by ERT at four months of age and then by HSCT at one year of age. Such a combined therapy has not yet been described in this disease, whereas in other MPS types, such as I or II, it led to improved transplantation conditions [7]. Additionally, this boy harbored three compound heterozygote missense mutations: one common substitution inherited from the father and linked to an attenuated phenotype [8] and two previously unknown mutations from the mother for whom we analyzed their functional impact on the GUSB protein.

### Case Description

The patient was the firstborn to non-consanguineous parents. No history of hydrops fetalis was recorded. Pregnancy and delivery were normal. The newborn was small for his age (birth weight, 2680 g; birth length, 44 cm; birth head circumference, 34 cm). He presented at birth a lymphedema, a coarse facies, a club foot (talipes equinovarus), and a slight hepatosplenomegaly. A thrombopenia was noticed and vacuolated leukocytes were found upon examination of the blood smear (Figure 1). A lysosomal storage disease was quickly suspected. The patient was born in a local hospital (Castres, France) and was referred to our university hospital (Toulouse) when he was nine days old.

## 2. Results

### 2.1. Biochemical Diagnosis and Characterization of Mutant GUSB Alleles

At 10 days of life, traces of dermatan sulfate (DS) were found in the patient’s urine (data not shown) along with a marked GUSB enzyme deficiency (<1% of control values) both in peripheral blood leukocytes and plasma (Figure 2). The diagnosis of MPS VII was then confirmed by Sanger sequencing of the *GUSB* gene, evidencing three missense variations in exon 3: c.526C>T (p.L176F), c.422A>C, and c.424C>T (p.E141A and p.H142Y) (numbered according to NM_000181.4). Analysis of the parents’ DNA demonstrated that the former was inherited from the father while the latter two originated from the mother. *In silico* analysis tools (PolyPhen2 and SIFT) predicted that the substitution of His142 has a deleterious impact on protein, as reported in Khan’s work [9]. As for the c.422A>C mutation, it is predicted as a likely pathogenic mutation by PolyPhen2 but tolerated by SIFT.

Because the c.422 and c.424 substitutions have never been described, we investigated the impact of these individual variations on both the expression and the catalytic activity of GUSB. To this purpose, HEK293T cells were transiently transfected with a plasmid containing either the wild-type (WT) sequence of the human *GUSB* cDNA or the following mutated sequences: c.526C>T, c.422A>C, or c.424C>T or the double mutation c.422A>C;424C>T. Constructs were prepared so as to express the C-terminal DYK tag. Cells were simultaneously transfected with a vector encoding the bacterial β-galactosidase for assessment of transfection efficiency and normalization of enzyme activity.

Western blot analysis showed similar levels of (tagged) GUSB expression in cells transfected with WT or mutant sequences (Figure 3A). These results indicate that the mutations did not significantly affect GUSB protein synthesis or stability.

Enzyme activity measurements showed a remarkable increase in GUSB activity in cells expressing the WT and mutant proteins as compared to non-transfected cells or cells transfected with the empty vector (Figure 3B). As compared to cells expressing the WT protein, a significantly reduced GUSB activity was observed in cells expressing the c.422A>C;424C>T double mutation (45% of WT activity) while a more modest decrease was found in cells expressing each single mutation (83% and 73% of WT activity). In addition, enzyme activity in cells overexpressing the c.526C>T mutation was also reduced (59% of WT activity). These results suggest that the mutations harbored by the patient are indeed responsible for a reduction of GUSB catalytic activity.

### 2.2. Patient’s History

When ERT was initiated at four months of age, the boy presented with a coarse face, hepatosplenomegaly, and bone deformities without cardiac involvement, respiratory involvement, nor corneal lesion. Dorsal flexion of the right foot (initial clubfoot deformation) was limited to 5°, while it was measured to 15° for the left foot. Other joints showed normal mobility. A slight axial hypotonia was observed, but brain magnetic resonance imaging (MRI) was normal. The patient had received rhGUSB (4 mg/kg) every two weeks during 14 months from 4 to 18 months of age. After one year under ERT, dysmorphia had progressively improved, hepatomegaly disappeared, spleen size decreased but remained abnormal, and no infections were noticed. Excretion of urinary HS and CS-DS under ERT, evaluated at weeks 2, 4, 6, 8 and 12, showed a decrease (Figure 4). Unilateral clubfoot deformation was managed by the Ponseti method, allowing a complete correction of the foot deformation. Other bone signs remained stable, except for a moderate spine stiffness, without severe involvement. ERT was well tolerated and no adverse effects were reported. At the age of one year, neurological evaluation was close to normal, except for global hypotonia and slight motor delay. Auditory evoked potentials showed bilateral partial deafness until the age of two.

To prevent or at least to delay potential neurological damage, a HSCT with unrelated donor umbilical cord blood was proposed and discussed with the parents. It was performed at the age of 13 months; the degree of HLA matching was optimal (6/6). The engraftment proceeded well and revealed a chimerism with 100% donor cells after 30 days and two years. The post-transplantation period was marked by a severe skin and gut graft-versus-host disease that was treated by cyclosporine, corticoids, and infliximab. Following this episode, the patient presented a severe anorexia, which required an enteral nutrition and resulted in a slowdown of growth rate (from −1 to −2 SD for age) and subsequently a slight psychomotor development stagnation. Enteral nutrition was stopped after specific management of anorexia, which allowed for restoration of normal growth rate and catch-up of developmental skills. The patient showed partial hypothyroidism, requiring substitution with L-Thyroxine. The last follow-up examination (at the age of four years) revealed stabilized growth curve on −2 SD for age, no hepatosplenomegaly, and no other organ involvement. Auditory evoked potential testing was normal, and the patient developed a normal language without any equipment or re-education. Psychomotor development was normal, with normal scholarship for age. Neuropsychological tests could not be performed because of parental refusal.

GUSB activity measurements revealed that, following transplantation, leukocyte activities reached normal levels. However, plasma activities remained very low (Figure 2). Urinary GAG levels were normal.

## 3. Discussion

The present case reports the first patient affected with MPS VII who was treated before the age of six months. ERT was initiated at four months of age, followed by HSCT at 13 months of age. Such an early treatment was possible as diagnosis was established at two weeks of age.

A high degree of phenotypic heterogeneity is described in Sly disease, which is also a very rare MPS, so much that it is difficult to predict clinical outcomes in very young patients. A recent survey of 53 postnatally diagnosed patients indicated that the onset of the disease often occurs on the first day of life, whereas the median age of diagnosis is 11 months [2]. About half of the patients had skeletal deformities and 30% of the patients presented with a cognitive impairment; no further details, however, on the neurological signs were given. The median age of survival of these postnatally diagnosed patients was 30 years. A higher residual GUSB enzyme activity in fibroblasts (but not in leukocytes or plasma) is associated with a longer survival [2]. At least 68 different *GUSB* mutations have been reported [10]. Tomatsu et al. described some genotype-phenotype correlations [8]. Our patient harbored the c.526C>T (p.Leu176Phe) mutation, which appears to be one of the most frequent mutations and is associated with an attenuated phenotype [8]. GUSB activity in c.526C>T-overexpressing cells was decreased but not significantly different from the WT-expressing cells, although the activity in patient’s cells was low. This discrepancy has already been described in another cell model (COS cells) and would result from the mass action [11]. The patient also harbored a double point mutation in exon 3 (c.422A>C;424C>T), which has never been described. A predictive software analysis showed that the c.424C>T mutation had a deleterious impact on protein [9]; uncertainty remained for the c422A>C variation. These two mutations are neither located in the catalytic site nor in the glycosylation sites and are not supposed to impair the trafficking of the enzyme (Figure 5). To study the impact of this double mutation, expression and activity of GUSB were measured in HEK293T cells expressing either each single mutation or the double mutation. We found that the impact of the double mutation was more deleterious on the catalytic properties of the enzyme than that of each single mutation, possibly as a result of enhanced steric hindrance due to adjacent mutations.

ERT, using a recombinant human GUSB enzyme, vestronidase alfa, had been developed a few years ago, but at the time of diagnosis of our infant case, it was not yet available in France and a compassionate use basis had to be required and was granted.

ERT is being routinely and successfully used in patients with various types of MPS [7]. Regarding MPS VII, ERT was first reported by Fox et al. in a 12-year-old boy with an advanced stage of MPS VII [6]. He presented with a very poor pulmonary function with frequent hospitalizations and great weakness. ERT improved his respiratory function and his quality of life. Recently, a pivotal phase 3 trial about vestronidase alfa, which enrolled 23 patients starting from the age of five months to 25 years, has shown a significant decrease of urinary CS-DS excretion after 24 weeks. The clinical efficacy, based on the improvement in a Multi Domain Responder Index (MDRI), including joint mobility, respiratory, neurological and vision tests, revealed variable results [5]. While a marked amelioration of organomegaly and fatigue was observed, only a relative improvement of the mobility, respiratory, and neurological functions was noted. In this present case, early ERT improved organomegaly and facial dysmorphia, and resulted in a decrease of urinary GAGs after 12 weeks of treatment (Figure 4). In addition, this treatment was quite well-tolerated as the patient did not present any side effects. However, this therapy requires weekly, lifelong intra-venous administrations.

The recombinant enzyme is directly infused into the bloodstream, allowing its delivery to all peripheral organs except the brain because of its inability to cross the blood brain barrier. Because the nature of the molecular *GUSB* defects identified in our patient did not allow any prediction as to the clinical outcome, and in order to prevent or limit any potential neurological impairment, an allogenic HSCT was discussed with the parents and performed. HSCT is based on the supply of deficient enzyme through non-deficient GUSB-producing hematopoietic cells and their derived cells like brain microglia [14]. HSCT as a treatment for Sly disease has already been reported [4,15,16,17,18] in nine patients (see Table 1). It appeared rather as a preventive treatment. When treated early and prior to the development of clinical signs, patients did not develop the classically associated signs of the disease, while in symptomatic patients HSCT did not reverse them but seemed to slow them. It should be noted that patients who presented very serious symptoms prior to transplantation died despite therapy [4]. Nevertheless, as observed in other MPS types, HSCT has some limitations, such as its efficiency in correcting heart valve infiltration and cornea or bone diseases [7]. Regarding the bone alterations (leg deformity and clubfoot) presented at birth by the patient, an orthopedic treatment was performed because ERT and HSCT were not expected to improve them. It is thought that the bone impairment occurs very early in life and that the treatment should begin prior to this damage. In a MPS VII dog model, a delay of secondary ossification has been identified with an incomplete cartilage-to-bone transition. This process normally happens in dogs between 9 and 14 days; very early in life [19]. Until now, no treatment has been really efficient on bone damage. Finally, HSCT also remains associated with a high morbidity and mortality [7], but it cannot be ignored that, when it is effective, HCST makes weekly hospitalizations for ERT treatment unnecessary and improves the quality of life of the patients.

The combination of ERT and HSCT has been associated with an improvement in pre- and post-transplantation conditions. ERT decreases GAG storage in organs and allows time to find a donor if need be [7]. Our patient did not present neurological, respiratory, nor cardiac complications and was thus considered a good candidate for this procedure, especially as he had been previously treated by ERT. ERT followed by HSCT had never been reported before in MPS VII patients but had already been used on MPS II patients, where the combination proved to be effective on growth [20] and life activities [21]. Likewise, in mice, a better therapeutic response was observed in the group treated by this combination compared to the animal group treated by ERT only [22].

Interestingly, after HSCT, restoration of the enzymatic activity was obtained in the patient’s leukocytes (reflecting a full chimerism) but surprisingly not in plasma (Figure 2). To our knowledge, this observation has only been reported once in a MPS II patient [23]. This raises the question regarding the source of the circulating lysosomal enzyme. Usually it is thought to originate from hematopoietic cells, but this hypothesis would not explain the very low plasmatic activity while blood leukocytes exhibited a normal activity. Another source could be suggested, like endothelial cells or peripheral organs, which do not derive from the engrafted cells. This raises the question of whether ERT should be continued after a successful HSCT. Indeed, the low plasmatic activity suggests that peripheral organs would not take up and would still lack normal enzyme, resulting in an elevated risk of GAG storage and tissue lesions. Long term complications in MPS type I after engraftment have been reported, such as obstructive sleep apnea syndrome [24]. This kind of symptom may indicate persistent GAG storage due to the lack of circulating normal enzyme.

In our case, no disease progression has been noticed since the procedure (with three years of hindsight). Nevertheless, a follow-up remains necessary.

Finally, it clearly appears that management must start very early to prevent complications (for brain and bone impairments in particular). While no unequivocal predictive biomarkers or prognosis factors exist so far, a cross-sectional analysis showed that a higher residual GUSB enzyme activity in fibroblasts and lower GAG excretion in urine were associated with longer survival [2].

## 4. Materials and Methods

### 4.1. Lysosomal Enzyme Assays

GUSB activity in cell lysates (leukocytes or HEK293T cells) and in plasma (obtained from EDTA blood) was determined using 4-methylumbelliferyl-β-D-glucuropyranoside as substrate (Sigma, Saint-Quentin Fallavier, France) at an acidic pH (0.5 M sodium acetate buffer, pH 4.5) [25,26]. *E. coli* β-galactosidase and endogenous neutral β-galactosidase activities in transfected HEK293T cells were determined using 4-methylumbelliferyl-β-D-galactopyranoside as substrate (Sigma). These assays were performed at a neutral pH (0.25 M Tris/HCl buffer, pH 7.0) and in the presence of 5 mM MgCl_2_. Alternatively, activities were assessed in the presence of EDTA (5 mM); the *E. coli* β-galactosidase activity was calculated by subtracting the value obtained in the presence of the chelator from that obtained in the presence of MgCl_2_. Protein concentration was determined using Bradford method.

### 4.2. Urinary GAG Analysis

GAG analysis is based on the quantification of total urinary GAGs and on the identification of individual GAGs after electrophoretic separation. Total GAGs were quantified using the dimethylmethylene blue method [27] or through the measurement of uronic acid or galactose contents. It should be noted that during the first 3 months of treatment, specific concentrations of urinary HS and CS-DS were measured by ARUP Laboratories (Salt Lake City, Utah). For their identification, GAGs were precipitated by cetylpyridinium chloride and separated by 1D-electrophoresis. After staining with Alcian blue, quantitative analysis was performed by densitometry [28].

### 4.3. DNA Studies

After receiving informed consent from the parents, mutational analysis was performed by Sanger sequencing of the genomic DNA isolated from peripheral blood leukocytes. DNA was extracted using the MagNA Pure 24 instrument (Roche Life Science, Meylan, France), and *GUSB* exonic sequences (including exon-intron boundaries) were PCR amplified using previously described primers [29]. The SIFT and PolyPhen2 softwares were used for evaluation of each single mutation.

### 4.4. Cell Lines and Transfections

Human embryonic kidney (HEK293T) cells were grown in a humidified 5% CO_2_ atmosphere at 37 °C in Dulbecco’s Modified Eagle Medium (DMEM 4.5 g/L glucose, Thermofisher^®^, Courtaboeuf, France) and 10% inactivated fetal calf serum. Cells were transfected using Lipofectamine^®^ 2000 (Thermofisher) with 0.5 or 1 µg of pCMV-LacZ plasmid and 0.5, 2, or 10 µg of the pcDNA3.1^+^/C-(K)-DYK plasmid either containing or not containing the wild-type (WT) or mutated human *GUSB* cDNA sequences. The latter were purchased from GenScript (Leiden, The Netherlands). After 48 h of incubation, cells were washed with PBS and harvested; cell pellets were frozen at −80 °C until use.

### 4.5. Western Blot Analysis

After cell lysis by a cell lysis buffer (Cell Signaling Technology/Ozyme, Saint-Cyr-L’École, France) containing 1 mM phenylmethylsulfonyl fluoride (Sigma) and a protease inhibitor cocktail (cOmplete, Roche), protein concentration was determined using the Bradford method. Then, 30µg protein extracts were loaded in 10% sodium dodecyl sulfate-polyacrylamide gel and blotted on a nitrocellulose membrane. Expression of GUSB was analyzed via the mouse anti-DYK antibody (diluted 1:1000 in Tris-buffered saline containing 0.1% Tween 20 (TBST) containing 5% bovine serum albumin) provided by Sigma; a mouse anti-β-actin antibody (dilution 1:4000 TBST, 5% milk) purchased from Cell Signaling Technology was used as a control for protein loading. Anti-mouse secondary antibodies (1:5000 TBST, 5% milk) were used. Proteins were detected using an ECL detection system (Pierce / Fisher Scientific, Illkirch, France).

### 4.6. Treatments and Assessments

At the time of diagnosis, no specific treatment was available in France for MPS VII, but ERT was being developed by Ultragenyx, using a recombinant human GUSB (rhGUSB, vestronidase alfa UX003). A compassionate use basis was granted, and ERT could be started at the age of four months, with specific authorization from the French ministry of health. The patient received rhGUSB (4mg/kg) every two weeks for 14 weeks (from 4 to 18 months of age).

Allo-stem cell allograft transplantation was performed at 13 months of age with unrelated donor umbilical cord blood. Conditioning regimen consisted of busulphan, cyclophosphamide, and anti-thymocyte globulins. He received a combination of cyclosporin and corticosteroids as graft-versus-host disease prophylaxis. ERT was maintained until six months after the transplantation and then stopped.

Efficacy of these treatments was evaluated by regular clinical examination (especially growth and psychomotor development) and by biological tests. Urinary GAGs were evaluated at baseline and weeks 2, 4, 6, 8, and 12 under ERT (by ARUP laboratories). In addition, enzymatic activities in leukocytes and plasma were assessed after HSCT. Safety was evaluated by assessing for any adverse events.

## 5. Conclusions

This case extends previous knowledge on the very rare disease MPS VII, and reports two novel likely pathogenic *GUSB* mutations. It highlights the benefit to start treatment very early thanks to the combination of ERT and HSCT, and may help define whether there is a critical therapeutic window in this form of MPS. It also emphasizes the need for in-depth characterization of GUSB mutants in an effort to study genotype-phenotype correlations. Finally, we question whether ERT should be continued after HSCT in order to provide a complete systemic correction of the pathology.

## Figures and Tables

**Figure 1 ijms-20-05345-f001:**
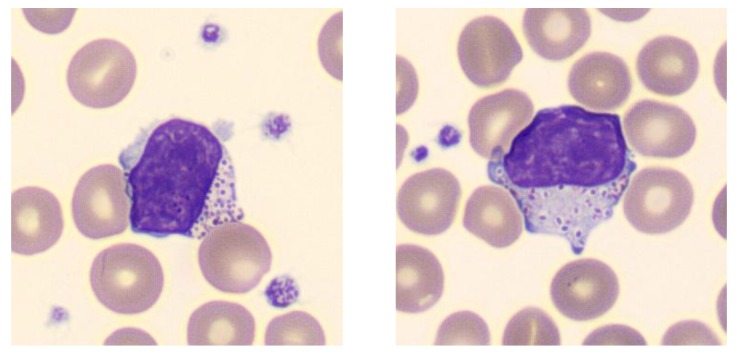
Patient’s peripheral blood lymphocytes showing Alder-like cytoplasmic inclusions (magnification ×100).

**Figure 2 ijms-20-05345-f002:**
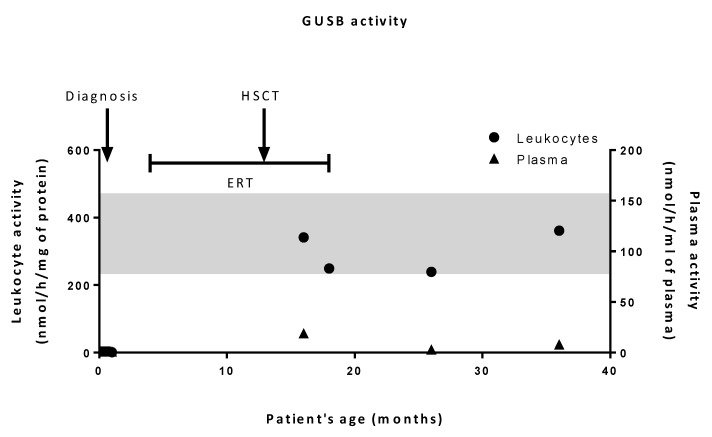
β-glucuronidase (GUSB) activities in patient’s leukocytes and plasma. GUSB activity was measured in patient’s plasma and leukocyte lysates at diagnosis and after hematopoietic stem cell transplantation (HSCT). Activities in leukocytes and plasma are expressed as nmol/h/mg protein and nmol/h/mL, respectively. The shaded band corresponds to the range of control values.

**Figure 3 ijms-20-05345-f003:**
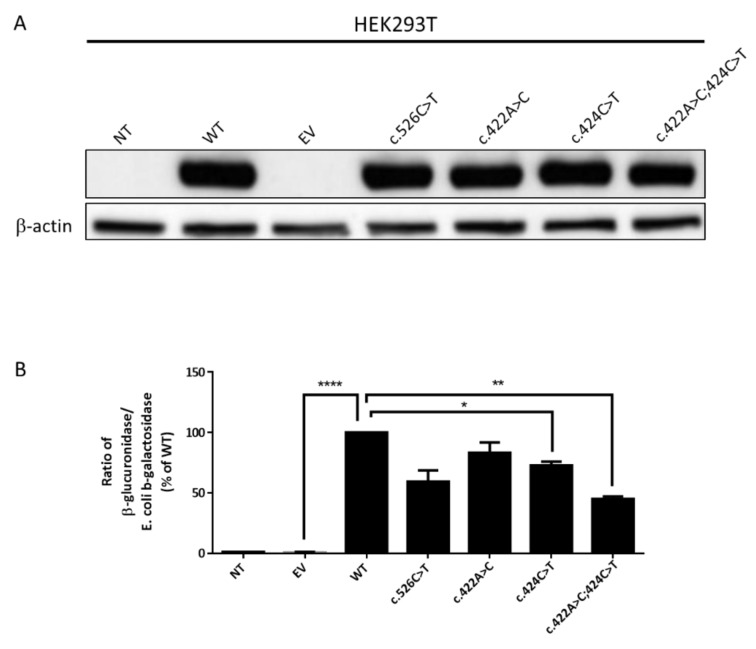
Effects of *GUSB* mutations on GUSB expression and enzyme activity. HEK293T cells were transfected or not (NT) with an empty vector (EV) or a vector containing the cDNA encoding either the wild-type (WT) or the indicated mutant GUSB. Forty-eight hours after transfection, cells were harvested and cell lysates were prepared. (**A**) Protein expression of GUSB. Lysates (30 µg protein) were analyzed for GUSB expression by Western-blot. The blot is representative of three independent experiments. (**B**) Enzymatic activity of GUSB in transfected HEK293T cells. GUSB and *Escherichia coli* β-galactosidase activities were measured in cell lysates. The ratio of GUSB/*E. coli* β-galactosidase activities (means +/− SEM of three independent experiments) is presented as a percentage of the values recorded in cells overexpressing the WT GUSB. The basal enzyme activity of GUSB in non-transfected cells averaged 30 nmol/h/mg protein. Asterisks indicate statistically significant difference: *, *p* < 0.05; **, *p* < 0.01; ****, *p* < 0.0001.

**Figure 4 ijms-20-05345-f004:**
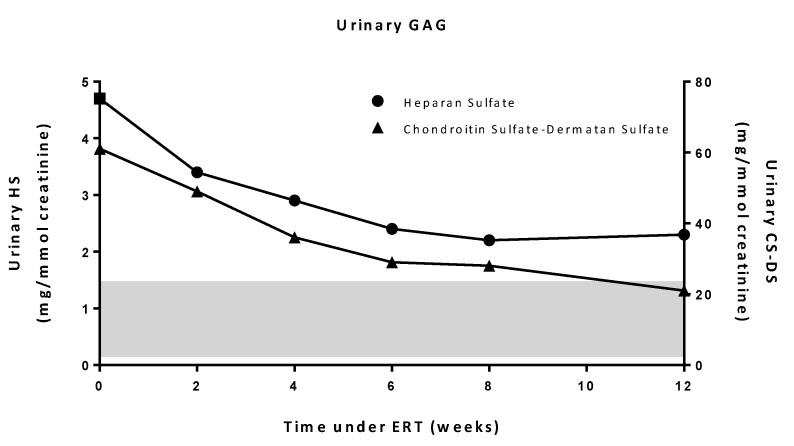
Levels of urinary heparan sulfate and chondroitin sulfate-dermatan sulfate under ERT. The value at time 0 corresponds to baseline levels prior to ERT. The shaded band corresponds to the range of control values.

**Figure 5 ijms-20-05345-f005:**
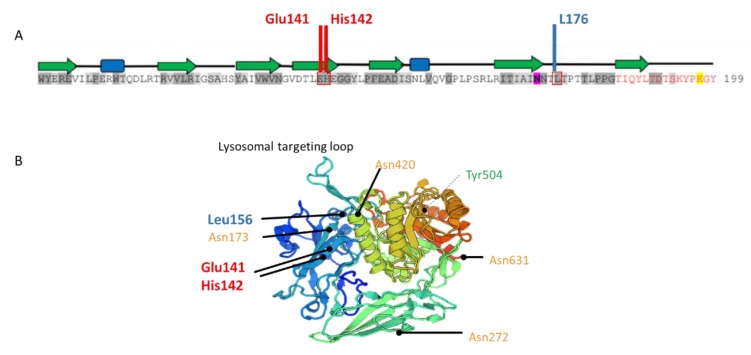
Localization of *GUSB* mutations. (**A**), Figure adapted from Hassan et al [12,13]. Amino acid sequence of human GUSB centered on the substitutions (in red boxes) carried by the present case. Residues shaded in dark and light grey correspond to degree of conservation between species. Residue highlighted in pink is a glycosylation site. On the top of sequence, in the secondary structure of GUSB, green arrows indicate β-strands and blue boxes correspond to α-helices. (**B**), Three-dimensional structure of GUSB monomer (https://swissmodel.expasy.org/). In orange, glycosylation sites; in green, active-site residue; in red and blue, the maternally (c.422A>C;424C>T) and paternally (c.526C>T) inherited mutated residues in the patient, respectively.

**Table 1 ijms-20-05345-t001:** Evolution of clinical features in patients who underwent bone marrow transplantation or HSCT. This table does not include the female patient described by Islam et al. [16], who was transplanted at the age of 11 months and survived at least 2 years, and for whom very scarce information is available.

References		Yamada et al., 1998 [15]	Montano et al., 2016 [4]					Sisinni et al., 2018 [18]	Furlan et al., 2018 [17]	Present Case
Age at diagnosis (months)		1 (girl)	22 (girl)	24 (boy)	4 (boy)	26 (boy)	0.5 (girl)	11 (girl)	1 (boy)	0.5 (boy)
Clinical signs	Hydrops fetalis	NM	NM	NM	NM	Yes	Yes	No	Yes	No
	Head, eyes, ear-nose-throat	Coarse face Mild deafness	NM	Coarse face	Coarse face	NM	Coarse face	Coarse face	Coarse face Bilateral severe hypoacusia	Coarse face Acute otitis media Mild deafness
	Cardio-respiratory	Recurrent respiratory infections Peripheral pulmonary stenosis Ventricular septal defect	NM	Recurrent respiratory infections Cardiac valve disease	Recurrent respiratory infections Cardiac valve disease	Breathing difficulty	Cardiac distress	Recurrent respiratory infections No cardiac disease	Respiratory distress recurrent infections	None
	Musculo-skeletal	Lumbar gibbus, bilateral femoral head hypoplasia Wheel-chair bound	Kyphosis and talipes equinovarus	NM	NM	NM	NM	Short trunk with pectus carinatum Scoliosis and kyphosis Talipes equinovarus Acetabular hip dysplasia Joint contractures and stiffness Genu valgum Clawed hands	Bilateral club-foot Pectus carinatum Gibbus Joint stiffness Dysostosis multiplex	Talipes equinovarus Kyphosis
	Thoraco-lumbar and abdominal	Hepatosplenomegaly Umbilical hernia	NM	NM	Hepato-splenomegaly	Hepatosplenomegaly	Hepatomegaly Inguinal hernia	Hepatomegaly Umbilical hernia	Hepatomegaly Bilateral inguinal hernias	Hepato-splenomegaly
	Neurological	Mental retardation (IQ: 50)	Normal intelligence	Mental retardation	Mental retardation	Mental retardation Development delay	NM	No alterations (IQ:100)	Axial hypotonia Hypertonia in limbs	Slight axial hypotonia
Age at transplantation (years)		12	2 and 4	7	NM	3	0.5	2 and 3.5	1.2	1.3
Evolution		Stabilization of symptoms Stop of recurrent infections Improved motor function (ability to walk short distances without assistance)	Moderate clinical manifestations Slow progression Persistent kyphosis and talipes equinovarus Normal intelligence	Deceased (from complications of the procedure)	Deceased (from complications of the procedure)	Moderate clinical phenotype Swallowing difficulties Recurrent respiratory infections Skeletal abnormalities Restrictive and obstructive airway disease	No clinical manifestations	Normal motor function Stabilization of musculoskeletal symptoms Improvement of coarse face and hepatomegaly Good respiratory function IQ: 109	Worsening respiratory disease Deceased at 25 months of age	Stabilized growth curve No hepatosplenomegaly Normal psychomotor development
Age at last follow-up (years)		14.5	NM			15	1.25	9		4

Abbreviation: NM, not mentioned.

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
