# Peer review of "First Report of a Patient with MPS Type VII, Due to Novel Mutations in GUSB, Who Underwent Enzyme Replacement and Then Hematopoietic Stem Cell Transplantation"

_ijms, 2019, doi:10.3390/ijms20215345_

Round 1

Reviewer 1 Report

The authors have presented interesting paper “First report of a patient with MPS type VII, due to novel mutations in GUSB, who underwent enzyme replacement and then hematopoietic stem cell transplantation.” for the International Joural of Molecular Sciences. The authers showed detailed analysis. There are some questions.

1 The author does not mention joint range of motion. If you were measuring, please mention.

2 What kind of neuropsychological test did the author perform regarding intellectual development assessment?

3 The author states in the auditory evoked potential test that the patient showed partial hearing loss on both sides. Please provide data on how much hearing loss you have had and describe the subsequent course.

Author Response

Responses to Reviewer 1

The authors have presented interesting paper “First report of a patient with MPS type VII, due to novel mutations in GUSB, who underwent enzyme replacement and then hematopoietic stem cell transplantation.” for the International Joural of Molecular Sciences. The authers showed detailed analysis. There are some questions.

 1 The author does not mention joint range of motion. If you were measuring, please mention.

Dorsal flexion of the right foot (initial clubfoot deformation) was limited to 5° while it was measured to 15° for the left foot. Other joints showed normal mobility.

This sentence has been added in the Patient’s history section, on page 4 line 125-126.

2 What kind of neuropsychological test did the author perform regarding intellectual development assessment?

Neuropsychological tests could not be performed because of parental refusal.

This sentence has been added in the Patient’s history section, on page 5 line 185.

3 The author states in the auditory evoked potential test that the patient showed partial hearing loss on both sides. Please provide data on how much hearing loss you have had and describe the subsequent course.

Auditory evoked potential testing was normal, and the patient developed a normal language without any equipment or re-education.

A correction has been made in the Patient’s history section, on page 4 line 136, and the above sentence has been added on page 5 line 183-184.

Reviewer 2 Report

Dubot and co-workers present the case report of a boy with MPS VII (Sly syndrome), who was treated with ERT followed by HSCT. To the best of my knowledge, this is the first report of such combined therapy for this disease. Simultaneously, it is also the first report of a Sly patient who got ERT treatment before the age of 6 months, highlighting (with its positive results over the child’s clinical course) the need for both an early diagnosis and treatment.

The authors further present a comprehensive review on the results of both ERT and HSCT in other MPS VII patients, which allow for a nice overview on the treatment’s effect in different subjects, at different ages. Also worth mentioning, the results they observe after HSCT for GUS activity (with a normalization of the activity in leukocytes, but extremely low levels in plasma) should also raise awareness on the possibility of persistent GAG storage in peripheral organs due to the lack of normal levels of circulating enzyme. This kind of reports is extremely important because it may contribute to a larger debate on the possible need of ERT even after HSCT and definitely highlight the need for a continuous and systematic follow-up of transplanted MPS patients.

Therefore, I definitely believe this manuscript is worth publishing. There are only a few minor points, which I feel should be corrected:

On page 3, line 100, the authors classify the enzymatic activity observed for c.526C>T as ‘slightly’ decreased. I would hardly use that term to classify the observed decrease (41%; vide figure 2), especially after they classify the 17 and 27% decreases seen for the single c.422A>C and c.424C>T as ‘moderate’. Thus, I would recommend the authors to skip the ‘slightly’ and rewrite the sentence in the following way: “In addition, enzyme activity in cells overexpressing the c.526C>T mutation was also reduced (59% of WT activity) as compared to WT cells”. On page 4, figure 2, there is no consistency on the use of the term “beta-glucoronidase” and the acronym “GUSB”. I would recommend the authors to skip the full name and stick to the initials, as they do throughout the overall manuscript. (The same happens in figure 1, actually). On page 5, line 147, I would recommend that the authors add a parenthesis statement with “(data not shown)” after saying that uGAG levels are normal. Otherwise we tend to look again at figure 1 (or 3), looking for them. On page 5, line 159, the authors state that there are 58 different mutations in GUSB and, although this could be truth in August, when they consulted the HGMD database, the current number is 68 (according to professional HGMD). So, I would recommend that they change the sentence (+ associated reference) accordingly (assessed on Oct 16, 2019). On page 7, line 215, there’s a reference missing for the canine MPS VII model On page 7, line 227, the authors may add reference to heart involvement after ERT + HSCT combined treatment, as it has also been described in reference 18. On page 7, line 245, there’s a question mark on “Do any predictive biomarkers or prognosis factors exist?” and, even though I feel this is a legitimate question, which could be discussed, I would either skip this sentence or further elaborate on the subject. On table 1, there are also a few corrections, which should be made: there’s a typo on the head, eyes, ear-nose-throat section for the first patient: a lost ‘k’ for the 4th patient described by Montano et al. (26, boy), the authors should add coarse face to the head, eyes, ear-nose-throat section, and inguinal hernia to the thoracolumbar and abdominal symptoms. For the 5th patient on the same paper, hepatomegaly should also be added to the list of symptoms. On the materials and methods section, the commercial suppliers for the anti-DYK and anti-beta actin antibodies are missing. also in this section, the authors use again the full ‘beta-glucoronidase’ term to refer to GUS enzyme assays. That should be corrected in accordance with the rest of the manuscript.

Author Response

Responses to Reviewer 2

Dubot and co-workers present the case report of a boy with MPS VII (Sly syndrome), who was treated with ERT followed by HSCT. To the best of my knowledge, this is the first report of such combined therapy for this disease. Simultaneously, it is also the first report of a Sly patient who got ERT treatment before the age of 6 months, highlighting (with its positive results over the child’s clinical course) the need for both an early diagnosis and treatment.

The authors further present a comprehensive review on the results of both ERT and HSCT in other MPS VII patients, which allow for a nice overview on the treatment’s effect in different subjects, at different ages. Also worth mentioning, the results they observe after HSCT for GUS activity (with a normalization of the activity in leukocytes, but extremely low levels in plasma) should also raise awareness on the possibility of persistent GAG storage in peripheral organs due to the lack of normal levels of circulating enzyme. This kind of reports is extremely important because it may contribute to a larger debate on the possible need of ERT even after HSCT and definitely highlight the need for a continuous and systematic follow-up of transplanted MPS patients.

Therefore, I definitely believe this manuscript is worth publishing. There are only a few minor points, which I feel should be corrected:

On page 3, line 100, the authors classify the enzymatic activity observed for c.526C>T as ‘slightly’ decreased. I would hardly use that term to classify the observed decrease (41%; vide figure 2), especially after they classify the 17 and 27% decreases seen for the single c.422A>C and c.424C>T as ‘moderate’. Thus, I would recommend the authors to skip the ‘slightly’ and rewrite the sentence in the following way: “In addition, enzyme activity in cells overexpressing the c.526C>T mutation was also reduced (59% of WT activity) as compared to WT cells”.

The sentence has been rewritten as suggested

On page 4, figure 2, there is no consistency on the use of the term “beta-glucoronidase” and the acronym “GUSB”. I would recommend the authors to skip the full name and stick to the initials, as they do throughout the overall manuscript. (The same happens in figure 1, actually).

Beta-glucuronidase is now named as GUSB throughout the manuscript (figures and captions included)

On page 5, line 147, I would recommend that the authors add a parenthesis statement with “(data not shown)” after saying that uGAG levels are normal. Otherwise we tend to look again at figure 1 (or 3), looking for them.

A parenthesis statement with “(data not shown)” has been added.

On page 5, line 159, the authors state that there are 58 different mutations in GUSB and, although this could be truth in August, when they consulted the HGMD database, the current number is 68 (according to professional HGMD). So, I would recommend that they change the sentence (+ associated reference) accordingly (assessed on Oct 16, 2019).

The sentence and the associated reference have been changed.

On page 7, line 215, there’s a reference missing for the canine MPS VII model

The reference 19 has been added

On page 7, line 227, the authors may add reference to heart involvement after ERT + HSCT combined treatment, as it has also been described in reference 18.

No further comment on heart has been added since the study reported by Tanaka et al (Mol. Genet. Metab. 2012) deals with HSCT(only)-treated MPS II patients, but not under a combined ERT-HSCT therapy. As a consequence, quotation of this study has been deleted. 

On page 7, line 245, there’s a question mark on “Do any predictive biomarkers or prognosis factors exist?” and, even though I feel this is a legitimate question, which could be discussed, I would either skip this sentence or further elaborate on the subject.

Rather than keeping the sentence with the question mark, the sentence has been modified.

On table 1, there are also a few corrections, which should be made: there’s a typo on the head, eyes, ear-nose-throat section for the first patient: a lost ‘k’ for the 4th patient described by Montano et al. (26, boy), the authors should add coarse face to the head, eyes, ear-nose-throat section, and inguinal hernia to the thoracolumbar and abdominal symptoms. For the 5th patient on the same paper, hepatomegaly should also be added to the list of symptoms.

Corrections have been made

On the materials and methods section, the commercial suppliers for the anti-DYK and anti-beta actin antibodies are missing. also in this section, the authors use again the full ‘beta-glucoronidase’ term to refer to GUS enzyme assays. That should be corrected in accordance with the rest of the manuscript.

The name of suppliers has been added, and the GUSB abbreviation indicated.